# Scientific text citation analysis using CNN features and ensemble learning model

**Khaled Alnowaiser**◉*

Department of Computer Engineering, College of Computer Engineering and Sciences, Prince Sattam Bin Abdulaziz University, Al-Kharj, Saudi Arabia

* k.alnowaiser@psau.edu.sa

**Data Availability Statement:** Data relevant to this paper are available from GitHub at https://github.com/nowaiserk/Plosone.

**Funding:** This study was supported by Prince Sattam bin Abdulaziz University in the form of a grant to KA [PSAU/2024/R/1445].

## Abstract

Citation illustrates the link between citing and cited documents. Different aspects of achievements like the journal's impact factor, author's ranking, and peers' judgment are analyzed using citations. However, citations are given the same weight for determining these important metrics. However academics contend that not all citations can ever have equal weight. Predominantly, such rankings are based on quantitative measures and the qualitative aspect is completely ignored. For a fair evaluation, qualitative evaluation of citations is needed in addition to quantitative ones. Many existing works that use qualitative evaluation consider binary class and categorize citations as important or unimportant. This study considers multi-class tasks for citation sentiments on imbalanced data and presents a novel framework for sentiment analysis in in-text citations of research articles. In the proposed technique, features are retrieved using a convolutional neural network (CNN), and classification is performed using a voting classifier that combines Logistic Regression (LR) and Stochastic Gradient Descent (SGD). The class imbalance problem is handled by the synthetic minority oversampling technique (SMOTE). Extensive experiments are performed in comparison with the proposed approach using SMOTE-generated data and machine learning models by term frequency (TF), and term frequency-inverse document frequency (TF-IDF) to evaluate the efficacy of the proposed approach for citation analysis. It is found that the proposed voting classifier using CNN features achieves an accuracy, precision, recall, and F1 score of 0.99 for all. This work not only advances the field of sentiment analysis in academic citations but also underscores the importance of incorporating qualitative aspects in evaluating the impact and sentiments conveyed through citations.

## Introduction

Scientific publications have increased substantially over the past decade and a large number of researchers are publishing their work in the form of articles and books worldwide. Consequently, many publications are available today with varying scientific quality and impact. Therefore, the necessity of reviewing and rating published scientific articles is in great demand. The literature has a wide range of evaluation standards for scientific writing. Nonetheless, the citation count is one of the most important evaluation indicators. The number of citations is frequently utilized to assess a paper's or researcher's impact [1–3]. In addition, it has served as

**Competing interests:** The authors have declared that no competing interests exist.

the foundation for many additional measures like h-index [4], impact factor, i-10 index [5] and other assessment parameters for researchers, conferences, journals, and research institutions [6, 7].

Scientific papers are linked to other cutting-edge scientific literature. In scientific contexts, the terms "citation" and "reference" are used to refer to the referenced work of other scholars [8]. Citing the work of other academics shows that scholars value their contributions and creates a connection between the referred study and the paper being cited [9]. Citations are essential for evaluating the quality of a work, however, the amount of the material used in citations makes it exceedingly challenging to extract information. Another difficulty in assessing the impact of publication is the expansion of scientific literature. Citation conveys both the effect of a publication and the authors' significance. Nonetheless, a quantitative metric is to count the citations that do not describe any qualitative expect of the citation. The number of times a research paper is cited by other researchers is known as the citation count, and it can be deceptive in judging the quality of a research study [10]. Because of the co-authorship, citations might be skewed and created with the purpose of increasing the reference count [11]. The paper is cited occasionally by the citing author only to identify short commands and suggestions for improvement. This sort of citation is also counted when calculating citation indexes [12]. Ranking methods cannot highlight highly effective research works by counting the number of citations only.

Citation sentiment analysis is a relatively understudied field of research. The majority of the citation in the research article explains the research findings without expressing an opinion; thus, the majority of citations are positive [12]. Some authors examined the sources in which criticism is frequently expressed in a courteous manner. Detection of the negative context of citations is a challenging task as their text is frequently implicit and concealed with contrasting terms. The positive context of citation involves its discussion with respect to design and evaluation criteria. Giving weight to the sentiments might help decide the text's polarity. In the past, researchers have worked on the detection of the sentiments from text using different approaches such as n-grams, dependency relations, lexical features, structure-based features, and many other state-of-the-art approaches [13]. Sentiment extraction from the citations is a challenging task as in most cases sentiments are hidden in citations and difficult to analyze. As per the studies, most citation sentiments are neutral because they are just used to support a phenomenon [14]. The study showed that the sentiment present in the citation text expresses the real feeling of the author towards another author's research. Predominantly, the sentiments of the author are explicitly hidden because they use an objective way to express the sentiments in the scientific papers [14].

The bibliometric measurements are the most important application of citation sentiment analysis. Citation sentiment analysis contributes to the improvement of bibliometric measurements. The prior method for determining an article's influence was to tally the number of times it was mentioned. Citation sentiment analysis, on the other hand, may be used to assign weights to each citation text based on the feelings of the citations [15]. Most of the time, the feelings indicated in the citation text are concealed, making it challenging to determine whether they are good, negative, or neutral [16]. In terms of people, it is simple for them to read the citation language and recognize the sentiment represented in it. But it becomes a complex and demanding undertaking when trying to train a model to automatically anticipate the polarity of feelings [17]. The citation text's emotion polarity appears to be neutral for the most part, with any concealed negative or positive attitudes. Many methods, including lexical analysis and feature extraction, can be utilized for hidden sentiment analysis. The following significant advances are made by this work using the machine learning technique for automated sentiment categorization of in-text sentiments.

- A machine learning-based framework is contrived to perform sentiment analysis for in-text citations of research articles. To improve the effectiveness of sentiment analysis, the proposed model combines a voting classifier with features generated using CNN. In addition, various machine learning models are utilized in this regard like decision tree (DT), Ada-Boost, logistic regression (LR), stochastic gradient descent (SGD), random forest (RF), extra tree classifier (ETC), support vector machine (SVM), and a voting classifier combining LR and SGD aka VC(LR+SGD).

- An effective text generation technique is proposed to handle class imbalance the problem for determining the citation text's sentiment into positive, negative, or neutral.

- The efficacy of the proposed approach is analyzed with and without using synthetic minority oversampling techniques (SMOTE). The performance of the Voting classifier in combination with CNN-based features is analyzed in terms of performance evaluation metrics.

The remainder part of the article is organized as follows. Several significant works that are connected to the current investigation are described in the Related Work section. The proposed technique, the dataset, and the models employed in the current work are all described in depth in the section Materials and Methods. Results and their discussion are covered in the section Results and Discussions, and the conclusion section is presented at the end of the paper.

## Related work

Several metrics have been established throughout the years to judge the quality of a research publication or authors. For example, the h-index is a significant metric for determining the author's rank and prominence [4]. In addition, impact factor and eigenvector are used for the same purpose [18, 19]. Nonetheless, these are quantitative indicators, and qualitative approaches to determine an article's rank have not been well-researched. Citation sentiment is a relatively new issue that may be used to overcome the constraints of quantitative techniques for analyzing the relevance of research publications. For example, Johson and Zang [20] designed a hybrid approach in which objective and subjective variables are integrated to assess the effect of a research publication. The study coupled the research paper's and author's impact factors with citation feelings for this goal. Later phases involve labeling each lemma and calculating its score with the SentiWordNet.

Some studies use citation sentiment with objective measures, while some methods consider only its sentiment. Ikram and Afzal [21] proposed a multilevel citation text analysis system by considering the aspects. With the help of the material immediately surrounding the reference, several elements are initially derived from the citation phrases. A linguistic rule-based technique is utilized to determine the polarity of these factors in sentiment analysis. The support vector machine is used with N-gram features to achieve the highest level of sentiment categorization accuracy. Similarly, Nguyen et al [22] presented a deep neural network for sentiment analysis of the text. LSTM approach is used with word embedding using word2vec while the data imbalance is dealt with using SMOTE. Results demonstrate that the proposed system outperforms the conventional SVM in terms of performance.

Athar [23] worked on context-based sentiment classification. The impact of window size on various techniques is examined through a variety of studies. The N-gram features' results demonstrate how the addition of contexts expands the vocabulary and influences performance. Ghosh and Shah [24] investigate the importance of the features for ranking a scientific article. The study performed the citing sentiment on the ACL paper collection. A few carefully chosen characteristics are used to train the models, including sentiment score, N-grams with positive

and negative polarity, self-citation, part of tags, and sentiment text. The findings demonstrate that Digging provides the greatest accuracy score of 80.61%.

The authors in [25] divided the citations into two categories: "influential" and "non-influential" The machine learning method (SVM) was used to help with this classification. A hundred research papers from the ACL Anthology were included in the dataset. There were five unique characteristics used: citation frequency, similarity, context, position, context, and other factors. A study [26] presented a content-based method for sentiment analysis in in-text citations, utilising binary classification to differentiate between significant and unimportant citations. The analysis' sentiment and cosine similarity scores were used as features in binary classification algorithms, filling a gap in the state-of-the-art literature where the ideal model for sentiment analysis was unknown. The work used automated sentiment analysis on extracted in-text citation material, tested several categorization models, and improved the understanding of sentiment analysis in citation settings.

Previously, numerous methods for identifying noteworthy citations were presented, including content-based, meta-data-based, and bibliographic-based approaches. However, even though the results were cutting-edge, they still needed to be improved. Authors in [27] used a two-module technique that included section-wise citation counting and sentiment analysis of citation sentences. The first module used Neural Network and Multiple Regression for automated weight assignment, while the second module used sentiment analysis for sentence-based categorization utilising Random Forest, Support Vector Machine, and Multi-layer Perceptron. Another research [28] proposed a new machine learning framework to distinguish important from non-important citations by analyzing syntactic and contextual information. Using three feature selection algorithms and three classifiers, the study identified key features for differentiation. Experimentation on two datasets demonstrated the framework's superior classification performance compared to contemporary research, highlighting the significance of both syntactic and contextual features in identifying important citations.

Several methods primarily concentrate on determining the citation's sentiments. For instance, Liu [9] proposed a system utilizing word2vec for citation text analysis. From ACL collections, sentence vectors are developed using word embedding. To calculate the polarity of the citation, the positive and negative polarity is used. SVM is employed with chosen characteristics to assess the citation's emotion. Results indicate that manual feature engineering performs better when trying to determine the polarity of a citation. Mercier et al [10] proposed a deep learning system called ImpactCite. The system is built on XLNet, which emphasizes emotion classification tasks and classification of intent, which reveals the citation's meaning. Results indicate that for micro-F1 and macro-1, the impact citations score yields scores of 88.13 and 88.93 respectively. and gives better performance than the present citation sentiment approaches.

A review of the available research indicates two key points: machine learning techniques outperform traditional approaches, and dataset imbalance has a considerable impact on the efficacy of such approaches. In this regard, this work employs a machine learning-based technique enhanced with CNN-based features using SMOTE to address the dataset imbalance issue.

## Materials and methods

This section explains the methodology and procedures employed in this study in detail. Fig 2 depicts the architecture of the suggested technique. Starting with data retrieval, the approach follows the generation of text by SMOTE to balance the dataset. Feature extraction is then

carried out that involves term frequency (TF), term frequency-inverse document frequency (TF-IDF), and CNN. The data is split for training and testing where the selected machine learning models are utilized for sentiment classification.

## Citation sentiment corpus

This study utilizes the 'citation sentiment corpus' taken from ACL Anthology Network [12]. The dataset contains 8,736 citation text annotated by human experts. Out of 8,736 citations records 829 are positive, 280 negative, and 7627 are neutral. These 8,736 citations are extracted from 194 research articles. After SMOTE augmentation/upsampling, we have 1000 citation records of each class to balance the dataset. The dataset comprises 'Source_Paper ID', 'Target_Paper ID', 'Citation_Text', and 'Sentiment'. The 'Source_Paper ID' is the citing paper's ID that represents the source of the text, 'Target_Paper ID' is the cited paper's ID, 'Citation_Text' is the original text that contains the citation while 'Sentiment' is the label of the target class and can be 'positive', 'negative', or 'neutral'. Table 1 shows a few example records from the collection.

## Machine learning classifiers

Supervised machine learning algorithms are extensively used to solve classification and regression problems [29]. Tree-based and regression-based algorithms are used in this study. This study used 8 different supervised algorithms to solve the classification problem. Table 2 provides implementation details about these machine learning models, and their hyperparameter settings for all of them. To find the best parameters, a method called grid search is used. This involves trying different values for each parameter within a specified range and evaluating how well the model works. Every parameter goes through the procedure, and the values that optimize the model's performance are selected at the conclusion.

**Decision tree.** The DT is a supervised machine learning model that learns discrete rules from data features to predict target variables [30, 31]. The main benefit of the DT is the

**Table 1. Example of different sentiments from the citation sentiment corpus.**

| Citation Text | Sentiment |
|---|---|
| "One of the most effective taggers based on a pure HMM is that developed at Xerox (Cutting et al., 1992)." | Positive |
| "Jing and McKeown (2000) have proposed a rule-based algorithm for sentence combination, but no results have been reported." | Negative |
| "To contrast, [Jing & McKeown, 2000] concentrated on analyzing human-written summaries in order to determine how professionals construct summaries." | Neutral |

**Table 2. Hyperparameter details of all machine learning models.**

| Classifier | Hyperparameter setting |
|---|---|
| RF | random_state = 50, max_depth = 20, n_estimators = 200 |
| AdaBoost | random_state = 50, max_depth = 20, n_estimators = 200 |
| SVC | C = 2:0, random_state = 500, kernel='linear' |
| LR | solver='saga', C = 3:0, max_iter = 100, penalty='l2' |
| SGD | Larning_rate='optimal', epsilon = 0.2 |
| DT | max_depth = 25, max_depth = 2−50 |
| ETC | random_state = 50, max_depth = 20, n_estimators = 200 |
| VC | Voting='hard' |

features subset that appears at different classification steps and decision rules. DT comprises different kinds of leaf nodes and various inside nodes having branches. Every leaf node denotes a target class while internal nodes denote features connected with branches to perform classification. The efficacy of the DT is based on how well-trained it is on the dataset.

**AdaBoost classifier.** AdaBoost from adaptive boosting is based on an ensemble learning classifier that utilizes the boosting method to train weak learners [31]. It combines many weak learners to recursively train them on the copies of the actual corpus, where every weak learner focuses on the outliers [32]. It is a metadata approach that uses the *N* number of weak learners and uses different assigned weights for training.

**Logistic regression.** LR is basically designed for binary classification but in this research, I have used LR One-vs-All (OvA) technique for classification. For each class, a binary classification model is trained to distinguish that class from all other classes. The final prediction is then based on the class with the highest probability among the individual classifiers. A statistical algorithm LR uses different variables to compute the final results. It is a regression-based model which estimates the class' probability. Therefore, it performs best for categorical data. To estimate the probability and ascertain the link between dependent and independent variables, LR employs the logistic function [33].

**Stochastic gradient classifier.** The working of SGD is similar to the LR and SVM. For the multi-class classification, SGD proves to be a robust classifier as it aggregates the various binary classifiers in a one-verses-all technique. SGD randomly selects the examples from the batch, so the hyperparameters of SGD need correct values to achieve precise results. It is highly sensitive towards scaling of features [34].

**Random forest.** RF comprises numerous decision trees which work separately to find the result while the decision on the ultimate outcome is made using the majority vote method. The outcome error rate is very less than other classifiers which is attributed to low correlation among trees [35]. Different split criteria are used for RF; the dataset is split on the basis of the Gini index which is the cost function. In RF, the bagging approach is utilized in which multiple classifiers are trained on bootstrapped data and used to minimize variance.

**Extra tree classifier.** ETC employs the meta estimator, which trains several weak learners using random samples from the dataset to enhance the prediction outcome [36, 37]. It is an ensemble model like RF widely utilized for classification problems. ETC differs from RF in the way of construction of trees in the forest. It uses actual data for training, unlike RF which uses bootstrap data samples. At every node, a tree uses *k* features of a random sample. Trees choose the best feature for splitting. These random feature samples lead toward the multiple de-correlated DTs.

**Support vector classifier.** SVC is basically designed for binary classification but in this research like LR, I have used SVC as One-vs-All (OvA) or One-vs-Rest (OVR) technique with linear kernel (OneVsRestClassifier(SVC(kernel='linear'))) for classification. For each class, a binary classification model is trained to distinguish that class from all other classes. The final prediction is then based on the class with the highest probability among the individual classifiers. The SVC, which Cortes and Vapnik first suggested, is a binary classification technique that may be expanded to handle issues with many classes [38]. The SVC is used to deal with multi-class classification problems. To deal with nonlinear classification, outlier detection, and regression support vector is a powerful technique. But the major drawback of SVC is that it does not give good results on small-sized corpus because it works on the cross-validation of data.

**Voting classifier.** Recently voting classifiers have shown better performance for many tasks than the traditional models. In a voting classifier, many classifiers can be added with respect to training time constraints, and a single regression model is used as a regression

model to calculate the voting outcomes. Every model forecasts the target label, and classifiers vote among themselves to select the target class label [40]. Soft and hard voting is used where soft voting considers the probability value of different classes from each classifier while hard voting considers classifiers' prediction with majority votes wins. This study combines LR and SGD as a voting classifiers.

## Feature extraction

The technique of finding meaningful features from the data for good and efficient training of the machine learning model is known as feature engineering. Techniques for feature engineering can help machine learning algorithms perform better. As a result of feature engineering, which separates the valuable feature from the raw data, the consistency and accuracy of the learning algorithm are improved. In this work, we used Vectorization (TF-IDF), prediction-based (TF), and SMOTE upsampling features. The strengths and weaknesses of these techniques are discussed in Table 3.

## Dealing with dataset imbalance

This study utilizes SMOTE and CNN-based features with a voting classifier to address the issue of an unbalanced dataset.

**Using synthetic minority oversampling technique.** SMOTE is a popular oversampling method for addressing the issue of unbalanced data. By leveraging Euclidian distance to generate random syntactic data of the minority class from its closest neighbors [41]. Because they are created using the original characteristics, newly produced instances resemble the original data pretty closely. To deal with high dimensional data SMOTE is not a good choice because it creates extra noise. A recent study uses the application of SMOTE to predict heart failure cases [42]. Machine Learning and SMOTE show reasonable results but still do not quite well to compete with deep learning models [43]. This study uses SMOTE to generate a new training dataset.

**Architecture of convolutional neural network for feature extraction.** In this work for scientific paper citation analysis, the deep learning model CNN is used as a feature extraction technique [44–46]. CNN is a widely used deep learning system mostly used for classification tasks. As a deep learning system has the capacity to extract features, the convoluted features are used for scientific paper citation sentiment analysis. The standard CNN model has four layers: an embedding, a convolutional, a pooling, and a flattening layer. For citation sentiment analysis, the first layer of CNN used is an embedding layer and it has an embedding size of 12, 000 and an output dimension of 100. The convolutional layer has 500 filters, a kernel size of 2,

**Table 3. Strength and weakness of feature representation technique.**

| Technique | Type | Strengths | Weaknesses |
|---|---|---|---|
| TF-IDF [38] | Vectorization technique | • Can find similarities between documents easily<br>• Calculate the words' occurrence in a document or whole corpus<br>• Weight is directly proportional to the words' frequency of documents and inversely proportional to words' frequency within documents.<br>• stopwords like a, the, is, but, etc. have no significance as compared to rare words. | • large vector size<br>• Position and co-occurrence has no importance<br>• Do not consider semantics and context.<br>• Sparsity problem<br>• Could not differ polysemy words or find similarities in synonyms. |
| TF [39] | Prediction based technique | • Count the words' occurrence within a document<br>• Consider word similarities<br>• Weight is directly related or proportional to the document's word frequency | • Do not consider semantics and context.<br>• Sparsity problem |

and a rectified linear unit (ReLU) as an activation function. The third layer is the max pooling layer; for the significant feature maps max pooling layer with 2 sizes is used from the output of the convolutional layer. The output is ultimately transformed into a 1D array using a flattened layer.

For example, a tuple set ($fs_i$, $tc_i$) is from the citation sentiment analysis dataset, where the $fs$ presents the feature set and $tc$ presents the target column, and $I$ show the index of the tuple. The embedding layer is used as a transformation tool to convert the training set into the needed input.

$$EL = embedding\_layer(Vs, Os, I) \tag{1}$$

$$EOs = EL(fs) \tag{2}$$

where $EL$ shows the embedding layer and $EO_s$ shows the embedding layers output which is the input of the convolutional layer. There are three different parameters for the $EL$: $V_s$ vocabulary size, I input length and Os is the output dimension.

In this study for citation sentiment analysis, the $EL$ size is set at 12, 000. It shows that the $EL$ can take the inputs from 0 to 12000. The input length is 42 and the output dimension $Os$ is set to 100. $EL$ processes all the input and gives the output for the CNN for additional processing. $EL$ output dimension is $EO_s$ = (None, 42, 100)

$$1D - Convs = CNN(F, Ks, AF) \leftarrow EOs \tag{3}$$

The convolutional layer output is extracted from the $EL$ output. CNN is implemented with the 500 filters, i.e., $F$ = 500, and a kernel size of 2. Utilizing the ReLU activation function, all negative values are set to zero while all other values are left unaltered.

$$f(x) = max(0, E)s \tag{4}$$

The map max pooling layer is used to extract features. For this purpose, a 2 pool is used. $Fmap$ shows the features after max-pooling, $Ps$ = 2 is the size of the pooling window and S-2 is the size of the stride. The last flattened layer is utilized for the data transformation. By using the above-mentioned steps we obtained the 251470 features for the training of the classifiers.

$$Cf = Fmap = \lfloor (1 - P_s)/S \rfloor + 1 \tag{5}$$

To convert the 2D data into 1D, a flattened layer is used. The machine learning models perform well on the 1D data, which is the primary driver behind this conversion. The aforementioned procedure is conducted during the training of the models.

## Proposed methodology

Ensemble models are becoming more prevalent and have led to greater accuracy and efficiency for classification tasks. By merging multiple classifiers, it is possible to enhance the performance beyond what individual models can achieve. In this work, an ensemble learning model is employed to enhance scientific paper citation sentiment classification. The proposed method extracts features from CNN and involves a voting classifier that unites the LR and SGD through the soft voting criterion. The ultimate output is determined by the class that receives the most votes. The proposed ensemble model, as outlined in Algorithm 1, operates as

follows:

$$\widehat{p} = argmax \sum_{i}^{n} LR_i, \sum_{i}^{n} SGD_i \qquad (6)$$

The prediction probabilities for each test sample are provided by $\sum_{i}^{n} LR_i$ and $\sum_{i}^{n} SGD_i$. These probabilities, as illustrated in Fig 1, pass through the soft voting criterion using the LR and SGD.

To demonstrate the capabilities of the proposed approach, let's consider an example. When a sample is evaluated by both the LR and SGD, it is assigned a probability score for each class. Suppose we have three classes, Class 1 (Positive), Class 2 (Negative), and Class 3 (Neutral), with likelihood scores of 0.4, 0.5, and 0.6, respectively, according to the LR model. For the same classes, the probability scores are 0.6, 0.7. and 0.9, respectively, according to the SGD model. Let P(x) be the probability score of x, where x belongs to the dataset's four classes. The probabilities for the three classes can be computed as follows:

$P(1) = (0.4 + 0.6)/2 = 0.50$

$P(2) = (0.5 + 0.7)/2 = 0.60$

$P(3) = (0.6 + 0.9)/2 = 0.75$

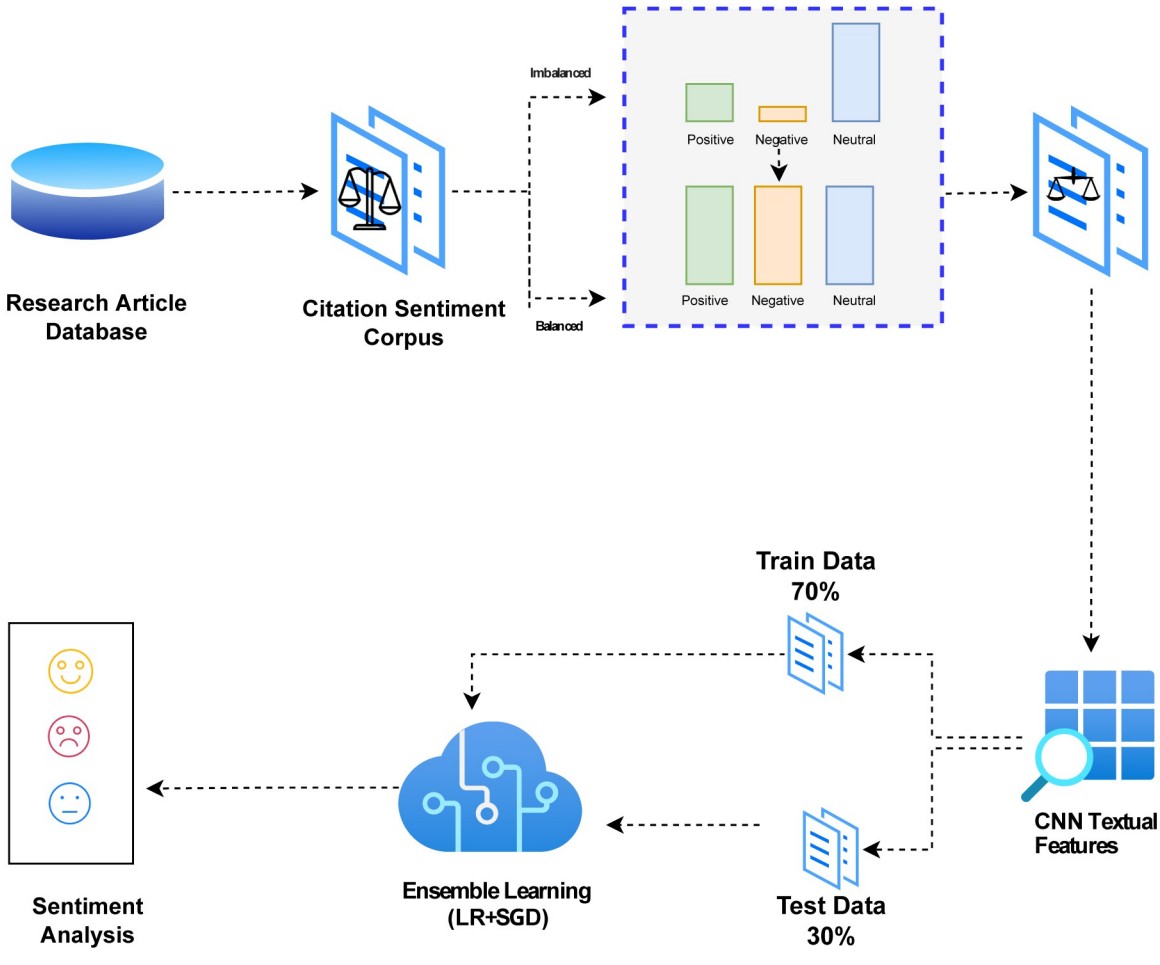

**Fig 1. Proposed architectural diagram.**

The ultimate prediction will be 3 (Neutral class), as it has the highest probability score, as shown below:

$$VC(LR + SGD) = argmax(g(x)) \qquad (7)$$

VC(LR+SGD) determines the final output by selecting the class with the highest average probability and adding the estimated probability from the two classifiers. The study's proposed citation sentiment analysis framework, which is illustrated in Fig 2 of the paper, involves the use of an ensemble model called VC(LR+SGD) that combines two machine learning models. This study employed the citation sentiment analysis dataset, which was acquired from the University of Cambridge Lab.

To assess the proposed model, the 'citation sentiment analysis dataset' is used in two stages. In the first stage, citation sentiment analysis is done using TF and TF-IDF features alone and with SMOTE. In the second stage of the experiments, the dataset is processed for machine-learning models using convolutional features. Two sections of data make up the whole, with 70% allocated for training and 30% reserved for testing. This approach, known as the training-

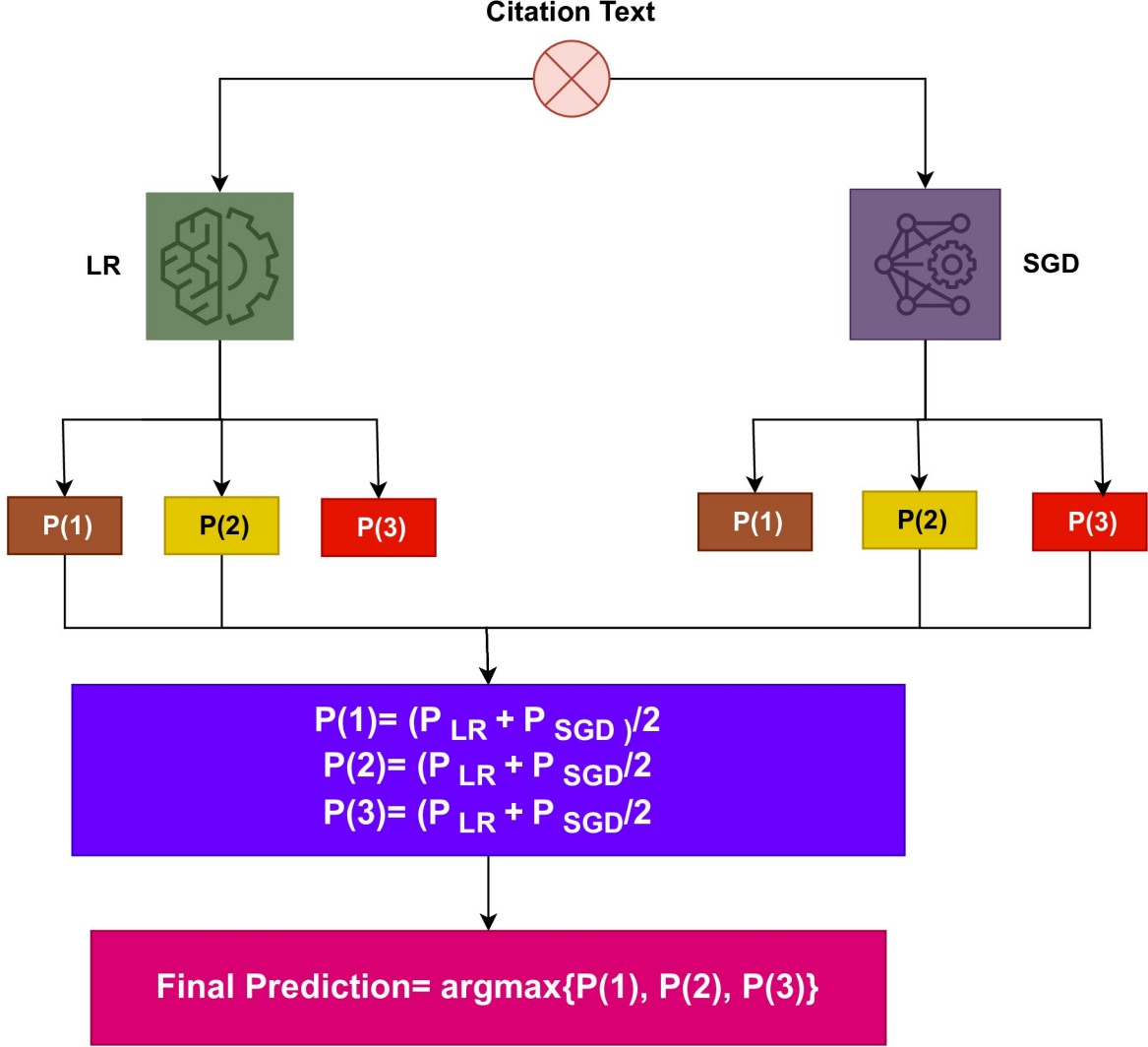

**Fig 2. Architecture of the proposed voting classifier (LR+SGD) model.**

testing split, is a common method in machine learning used to assess the accuracy of the model on new and unseen data.

**Algorithm 1** Ensembling LR and SGD.

```
Input: input data (x, y)ᵢ₌₁ᴺ
M_LR = Trained LR
M_SGD = Trained SGD
  for i = 1 to M do
    if M_LR ≠ 0 & M_SGD ≠ 0 & training_set ≠ 0 then
    P_LR₁ = M_LR.probability(class1))
    P_LR₂ = M_LR.probability(class2))
    P_LR₂ = M_LR.probability(class3))
    P_SGD₁ = M_SGD.probability(class1))
    P_SGD₂ = M_SGD.probability(class2))
    P_SGD₂ = M_SGD.probability(class3))
    Decision function = max(1/n Σ_classifier (Avg(P_LR₁, P_SGD₁),
    Avg(P_LR₂, P_SGD₂, Avg(P_LR₃, P_SGD₃))
  end if
  return final label p̂
  end for
```

# Results

The performance of several classifiers is evaluated using different evaluation parameters for citation text analysis. This work uses accuracy, precision, recall, and F1 score as the evaluation metrics. For the implementation of the machine learning algorithm, the sci-kit-learn library and NLTK have been utilized in Python. For training and testing, the data are divided into 0.7:0.3 ratios.

## Performance of classifiers using TF without SMOTE

The efficiency of the classifiers has been compared using TF without SMOTE to analyze citation text. The voting classifier achieves the greatest accuracy of 0.9122, according to the results shown in Table 4. SVC received a score of 0.8961, the second-highest for accuracy. LR, RF, and ETC show almost similar results in terms of precision, recall, and F1 score for citation sentiment analysis. However, Across all models, DT has the poorest outcomes using TF with a 0.8473 accuracy score.

## Performance of classifiers using TF with SMOTE

Supervised machine learning models have been evaluated using TF features with SMOTE. From Table 5, it can be clearly observed that combining TF and SMOTE substantially

**Table 4. Classification result of classifiers models using TF without SMOTE.**

| Models | Acc. | Prec. | Recall | F1 |
|---|---|---|---|---|
| DT | 0.8473 | 0.84 | 0.85 | 0.84 |
| AdaBoost | 0.8752 | 0.85 | 0.88 | 0.85 |
| LR | 0.8714 | 0.84 | 0.87 | 0.82 |
| SGD | 0.8870 | 0.87 | 0.89 | 0.86 |
| RF | 0.8760 | 0.84 | 0.88 | 0.84 |
| ETC | 0.8775 | 0.85 | 0.88 | 0.84 |
| SVC | 0.8961 | 0.87 | 0.89 | 0.87 |
| VC | 0.9122 | 0.90 | 0.90 | 0.90 |

**Table 5. Classification result classifiers using TF with SMOTE.**

| Models | Acc. | Prec. | Recall | F1 |
|---|---|---|---|---|
| DT | 0.9010 | 0.90 | 0.90 | 0.90 |
| AdaBoost | 0.8361 | 0.84 | 0.79 | 0.82 |
| LR | 0.9388 | 0.94 | 0.94 | 0.94 |
| SGD | 0.9361 | 0.96 | 0.96 | 0.96 |
| RF | 0.9729 | 0.98 | 0.96 | 0.97 |
| ETC | 0.8444 | 0.84 | 0.84 | 0.84 |
| SVC | 0.9669 | 0.97 | 0.97 | 0.97 |
| VC | 0.8667 | 0.86 | 0.87 | 0.86 |

**Table 6. Classification result of classifiers using TF-IDF without SMOTE.**

| Models | Acc. | Prec. | Recall | F1 |
|---|---|---|---|---|
| DT | 0.8473 | 0.84 | 0.85 | 0.84 |
| AdaBoost | 0.8752 | 0.85 | 0.88 | 0.85 |
| LR | 0.8714 | 0.84 | 0.87 | 0.82 |
| SGD | 0.8870 | 0.87 | 0.89 | 0.86 |
| RF | 0.8760 | 0.84 | 0.88 | 0.84 |
| ETC | 0.8775 | 0.85 | 0.88 | 0.84 |
| SVC | 0.8961 | 0.87 | 0.89 | 0.87 |
| VC | 0.9122 | 0.90 | 0.90 | 0.90 |

improved the performance of all classifiers for sentiment analysis of citation text. The best model continues to be RF, which achieves accuracy scores of 0.9729, precision scores of 0.98, recall scores of 0.96, and F1 scores of 0.97. DT, LR, SGD, RF, and SVC show accuracy higher than 0.90 for all evaluation matrices. AdaBoost performs the poorest, with accuracy, precision, recall, and F1 score values of 0.8361, 0.84, 0.79, and 0.82 respectively. The performance of VC is poor in this case because both classifiers have the same way of learning feature patterns (linear feature capturing). If the features are quite similar to each other, than there are more chances that both classifiers makes similar types of errors and performs poor like in our case.

## Performance of classifiers using TF-IDF without SMOTE

Without utilizing SMOTE, the outcomes of classifiers that use the feature extraction method TF-IDF are compared. The accuracy, precision, recall, and F1 score comparison of classifiers employing TF-IDF is shown in Table 6. It is observed that the voting classifier outperforms other models with an accuracy score of 0.9122 and 0.90 values each for precision, recall, and F1. SVC shows a marginally lower performance with a 0.8961 accuracy score, 0.87 precision, 0.89 recall, and 0.87 F1 score. For citation sentiment analysis, RF and ETC produce comparable findings with accuracy scores of 0.8760 and 0.8775, respectively.

## Performance of classifiers using TF-IDF with SMOTE

After using SMOTE, the performance of the models is also assessed using the TF-IDF. Results given in Table 7 provide the comparison of classifiers using TF-IDF with SMOTE balanced dataset to analyze sentiments of citation text. It is evident that classifiers that use TF-IDF with SMOTE perform better than classifiers that use TF-IDF without SMOTE. With accuracy scores of 0.9729, precision, of 0.98, and F1 scores of 0.96, RF had the best outcomes. All models

**Table 7. Classification result classifiers using TF-IDF with SMOTE.**

| Models | Acc. | Prec. | Recall | F1 |
|---|---|---|---|---|
| DT | 0.9010 | 0.90 | 0.90 | 0.90 |
| AdaBoost | 0.8361 | 0.84 | 0.79 | 0.82 |
| LR | 0.9388 | 0.94 | 0.94 | 0.94 |
| SGD | 0.9361 | 0.96 | 0.96 | 0.96 |
| RF | 0.9729 | 0.98 | 0.96 | 0.97 |
| ETC | 0.8444 | 0.84 | 0.84 | 0.84 |
| SVC | 0.9669 | 0.97 | 0.97 | 0.97 |
| VC | 0.8667 | 0.86 | 0.87 | 0.86 |

**Table 8. Classification results of machine learning models using CNN features with.**

| Models | Acc. | Prec. | Recall | F1 |
|---|---|---|---|---|
| DT | 0.9610 | 0.95 | 0.96 | 0.95 |
| AdaBoost | 0.9316 | 0.91 | 0.93 | 0.92 |
| LR | 0.9765 | 0.94 | 0.94 | 0.94 |
| SGD | 0.9661 | 0.96 | 0.96 | 0.96 |
| RF | 0.9814 | 0.97 | 0.94 | 0.96 |
| ETC | 0.9602 | 0.94 | 0.97 | 0.96 |
| SVM | 0.9618 | 0.97 | 0.97 | 0.97 |
| **VC(LR+RF)** | **0.9868** | **0.97** | **0.98** | **0.98** |
| **VC(SGD+RF)** | **0.9894** | **0.98** | **0.98** | **0.98** |
| **VC** | **0.9922** | **0.99** | **0.99** | **0.99** |

have shown significant improvement in classification accuracy after applying SMOTE. SVC achieved values higher than 0.96 in terms of all evaluation measures. Several factors could contribute to this decrease in the performance of the VC. First, the synthetic samples generated by SMOTE may introduce noise or artificial patterns that do not align well with the true distribution of the data. This can adversely impact the decision boundaries learned by the VC, resulting in a decrease in overall performance. Second, the specific characteristics of the VC, which combines multiple base classifiers, may interact with the synthetic samples in a way that hinders its ability to generalize to the true underlying data distribution.

## Performance of classifiers using CNN features

Finally, the result of classifiers is compared using TF-IDF with CNN to analyze citation text. Results are given in Table 8 which shows that all models have achieved improved and better results as compared to the results obtained by applying TF. DT, LR, SGD, RF, ETC, SVC, and voting classifiers achieved higher than 0.94 accuracy scores. RF and SVC have shown similar precision with a 0.97 score which is the second-highest. SVC has achieved a 0.97 value of recall and F1 score. However, ETC has achieved the highest results with a 0.9922 accuracy, 0.99 precision, recall, and F1 score.

## Results of cross-validation

To further verify the effectiveness of the proposed strategy, a 5-fold cross-validation is also conducted. The findings are shown in Table 9. Observably, the suggested model has an average

**Table 9. Significance of proposed methodology using k-fold validation.**

| K-folds | Acc. | Prec. | Recall | F1 |
|---|---|---|---|---|
| 1st-Fold | 0.989 | 0.994 | 0.995 | 0.994 |
| 2nd-Fold | 0.992 | 0.996 | 0.996 | 0.996 |
| 3rd-Fold | 0.997 | 0.993 | 0.998 | 0.995 |
| 4th-Fold | 0.998 | 0.994 | 0.997 | 0.996 |
| 5th-Fold | 0.999 | 0.999 | 0.999 | 0.999 |
| **Average** | **0.995** | **0.995** | **0.997** | **0.996** |

accuracy of 0.995 while its average precision, recall, and F1 score are 0.995, 0.997, and 0.996, respectively.

## Discussions

The comparison of the classifiers using TF and TF-IDF with SMOTE and text synthesis using CNN is shown in Fig 3. When employed with SMOTE balanced data, it demonstrates the Voting classifier's superiority over all other classifiers. When combined with TF-IDF and SMOTE, RF outperformed every combination of TF-IDF with classifiers, achieving 0.9829 accuracies, 0.98 precision, 0.96 recall, and 0.97 F1 score. Lastly, the findings show that the right mix of feature extraction approaches is essential for supervised machine learning models to be effective. For the analysis of the imbalanced text data, data balancing technique such as SMOTE improves the performance of the classifiers. Tree-based algorithm RF presents better results when trained on SMOTE-balanced data using TF-IDF features for sentiment analysis of citation text. If RF hyperparameter tweaking is done correctly, variance drops, and bias increases based on trees. Results reveal that the statistical technique (SMOTE) used to balance the data before training improves the performance of the classifiers. It seems that classifiers are not trained well when classes are imbalanced in

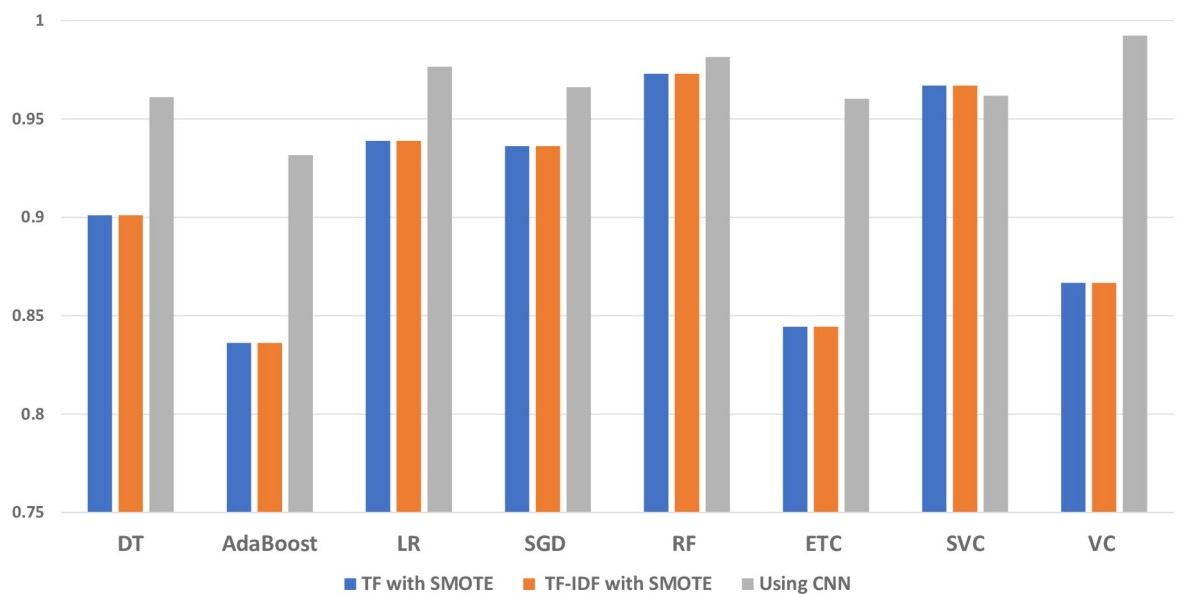

**Fig 3. Accuracy comparison of classifiers.**

text classification. When used properly with a features extraction technique called TF-IDF for sentiment analysis of citation text, tree-based models are more generalized and outperform other models.

Though SMOTE is a very useful technique in improving the performance of the models with class imbalance problems, it also has some limitations. For example, most of the generated synthetic samples are in one direction and complicate the decision process for the classifiers. SMOTE also creates a large number of noisy data samples and adds noise to the dataset. Table 10 presents the training and testing accuracy comparison of TF and TF-IDF features with SMOTE. The models' training and testing accuracy values may be shown to differ significantly from one another. DT, RF, ETC, and voting classifiers have shown 100% training accuracy but lower values for testing accuracy when trained on TF with SMOTE. ETC, SVC, and voting classifier have shown 100% training accuracy while a lower value of testing accuracy is observed when trained on TF-IDF with SMOTE. Hence results prove the overfitting of the models.

## Comparative analysis with cutting-edge methods

The proposed model's performance is compared with state-of-the-art research work based on feature engineering and learning models [47]. The reason for selecting this research for comparison is that this research work also utilized 4 different types of feature engineering for optimizing the performance of citation sentiment analysis. The same interest is ours but with a new feature engineering technique (CNN features) that outperformed the all techniques used in [47] research work. Table 11 displays the results of all models, which reveal that the voting classifier using CNN features yields the highest performance than other models. In [47], CNN performed better with 0.93 accuracies, 0.94 precision, 0.96 recall, and 0.95 F1 score. On the hand, the proposed approach used in this paper is VC(LR+SGD) using CNN features have shown superior performance with 0.99 accuracy, 0.99 precision, 0.99 recall, and 0.99 F1 score.

## Conclusions

Citation sentiment analysis has emerged as an attractive solution to complement the limitation of quantitative measures like citation count, h-index, etc. Analyzing the positive, negative, and neutral sentiments of the citing authors helps to determine the importance of a research article. Current citation sentiment analysis approaches face two challenges: low accuracy and dataset imbalance. This research endeavors to resolve the issue of dataset imbalance by using SMOTE and improved accuracy results using CNN for feature extraction and a voting classifier for classifying sentiments of citation text.

**Table 10. Training testing accuracy result of TF and TF-IDF features with SMOTE.**

| Models | TF Features with SMOTE | | TF-IDF Features with SMOTE | |
|---|---|---|---|---|
| | Training Acc. | Testing Acc. | Training Acc. | Testing Acc. |
| DT | 1.00 | 0.90 | 0.98 | 0.90 |
| AdaBoost | 0.99 | 0.83 | 0.98 | 0.83 |
| LR | 0.98 | 0.93 | 0.99 | 0.93 |
| SGD | 0.98 | 0.93 | 0.99 | 0.93 |
| RF | 1.00 | 0.97 | 1.00 | 0.97 |
| ETC | 1.00 | 0.84 | 1.00 | 0.84 |
| SVM | 0.99 | 0.96 | 1.00 | 0.96 |
| Voting Classifier | 1.00 | 0.86 | 0.99 | 0.86 |

**Table 11. Classification results of classifiers using fastText.**

| Model | Accuracy | Precision | Recall | F1 Score |
|---|---|---|---|---|
| Classification results of classifiers using fastText | | | | |
| CNN | 0.89 | 0.87 | 0.85 | 0.86 |
| LSTM | 0.86 | 0.70 | 0.74 | 0.72 |
| RF | 0.86 | 0.79 | 0.86 | 0.81 |
| SGD | 0.86 | 0.76 | 0.87 | 0.81 |
| LR | 0.86 | 0.76 | 0.87 | 0.81 |
| Classification results of classifiers using fastText subword | | | | |
| CNN | 0.87 | 0.85 | 0.82 | 0.83 |
| LSTM | 0.87 | 0.84 | 0.80 | 0.81 |
| RF | 0.87 | 0.82 | 0.87 | 0.83 |
| SGD | 0.87 | 0.76 | 0.87 | 0.82 |
| LR | 0.87 | 0.76 | 0.87 | 0.81 |
| Classification results of classifiers using GLOVE | | | | |
| CNN | 0.86 | 0.88 | 0.89 | 0.88 |
| LSTM | 0.84 | 0.78 | 0.74 | 0.76 |
| RF | 0.85 | 0.81 | 0.86 | 0.80 |
| SGD | 0.85 | 0.73 | 0.86 | 0.79 |
| LR | 0.85 | 0.76 | 0.85 | 0.79 |
| Classification results of proposed model | | | | |
| CNN | 0.93 | 0.94 | 0.96 | 0.95 |
| LSTM | 0.89 | 0.91 | 0.93 | 0.92 |
| RF | 0.87 | 0.89 | 0.92 | 0.90 |
| SGD | 0.88 | 0.91 | 0.91 | 0.91 |
| LR | 0.91 | 0.90 | 0.88 | 0.89 |
| Classification results of proposed model | | | | |
| **VC (LR+SGD)** | **0.99** | **0.99** | **0.99** | **0.99** |

Extensive experiments are performed using TF, TF-IDF, and CNN features with SMOTE to analyze the performance of the model's accuracy. Experimental results show that VC(LR+SGD) is the best-performing model for citation text classification when it is applied to the CNN-extracted features. The proposed approach achieves the highest accuracy score of 0.9922 while the values for precision, recall, and F1 score is 0.99 each. It is observed that when trained on SMOTE-generated data, models exhibit overfitting, but CNN features do not exhibit this issue. The future work of this research work is the ensemble of machine-deep learning models with a feature fusion of hand-crafted and word embedding techniques. The second future work direction is to make use of hybrid feature selection like merging two different word embedding with PCA.

## Author Contributions

**Conceptualization:** Khaled Alnowaiser.

**Data curation:** Khaled Alnowaiser.

**Formal analysis:** Khaled Alnowaiser.

**Funding acquisition:** Khaled Alnowaiser.

**Investigation:** Khaled Alnowaiser.

**Methodology:** Khaled Alnowaiser.

**Project administration:** Khaled Alnowaiser.

**Resources:** Khaled Alnowaiser.

**Software:** Khaled Alnowaiser.

**Supervision:** Khaled Alnowaiser.

**Validation:** Khaled Alnowaiser.

**Visualization:** Khaled Alnowaiser.

**Writing – original draft:** Khaled Alnowaiser.

**Writing – review & editing:** Khaled Alnowaiser.

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
