## [Decision Letter · Decision Letter 0]

10 Oct 2023

PONE-D-23-24549Scientific Text Citation Analysis Using CNN Features and Ensemble Learning ModelPLOS ONE

Dear Dr. Alnowaiser,

Thank you for submitting your manuscript to PLOS ONE. After careful consideration, we feel that it has merit but does not fully meet PLOS ONE’s publication criteria as it currently stands. Therefore, we invite you to submit a revised version of the manuscript that addresses the points raised during the review process.

We look forward to receiving your revised manuscript.

Kind regards,

Mohamed Hammad, Ph.D.

Academic Editor

PLOS ONE

Journal Requirements:

Reviewers' comments:

Reviewer's Responses to Questions

**Comments to the Author**

1. Is the manuscript technically sound, and do the data support the conclusions?

Reviewer #1: Yes

Reviewer #2: Partly

Reviewer #3: Yes

2. Has the statistical analysis been performed appropriately and rigorously? 

Reviewer #1: Yes

Reviewer #2: N/A

Reviewer #3: Yes

3. Have the authors made all data underlying the findings in their manuscript fully available?

Reviewer #1: Yes

Reviewer #2: Yes

Reviewer #3: Yes

4. Is the manuscript presented in an intelligible fashion and written in standard English?

Reviewer #1: Yes

Reviewer #2: Yes

Reviewer #3: Yes

5. Review Comments to the Author

Reviewer #1: This study considers multi-class tasks for citation sentiments on imbalanced data. In the proposed technique, features are retrieved using a convolutional neural network (CNN), and classification is performed using a voting classifier that combines Logistic Regression (LR) and Stochastic Gradient Descent (SGD). Extensive experiments are performed in comparison with the proposed approach using synthetic minority oversampling technique (SMOTE) generated data and machine learning models by term frequency (TF), and term frequency-inverse document frequency (TF-IDF) to evaluate the efficacy of the proposed approach for citation analysis.

Minor comments:

1. The Abstract must include the statistical findings and the main contribution.

2. The Introduction need more discussion about the research problem

3. Add more related works

4. Methodology as figure its better

Reviewer #2: This study investigates use of various machine learning models as well as use of CNN-based feature extraction and SMOTE-based data augmentation methods in sentiment prediction for the scientific citations.

The manuscript have some flaws and is not recommended for publication in PLOS One. There is not much novelty found in the presented work.

There are some incomplete sentences and some statements do not make much sense; e.g. in the abstract, the proposed method has been claimed to ‘outperformed’ (line-17) but there was no specification found what models were performing less than the proposed one).

Two different classifier were chosen to perform a voting classifier; this is not a standard practice; normally an odd number (e.g. 3, 5, ..) of classifiers would be chosen for a voting classifier model. No justification has been provided for choice of two classifiers (LR and SGD) for the voting classifier; how those two classifiers were selected among many others? Yet, the combination of these models has been already shown in a previous work (which has been already cited by the authors [44]), there is not much novelty found in the work presented in this paper.

The total sample size was given (line 158) however exact sample size for each class was not explicitly provided; similarly the readers are given the training:testing split ratio of 70%:30% but we do not know how many samples in each class and in both training and testing set exist. Similarly, after data augmentation (i.e. using SMOTE), the number of the samples exist in each class was not specified.

No discussion provided about why the voting classifier was one of the worst performing ones when the SMOTE was employed to balance the classes’ sample sizes (e.g. Tables 4 and 6).

No comments provided on why TF and TF-IDF use have been concluded with the exact same performance values for all the classifiers (in both cases when SMOTE is employed and not employed)?

The bars in Fig-3 that represent the accuracy of the VC do not look quite correct (especially the final 2 bars representing TF+CNN features and TF-IDF+CNN features) according to the text and tables provided in the manuscript

There is not any novelty in this draft when compared to the cited work [44] (https://doi.org/10.3390/app12063203).

The results shown in the Tables 3 to 7 were expected to be shown on the test set only (the results for the training could have been given as a suppl data or could have been provided explicitly in the main document) but the correct comparison between the models should be performed on the test set. As a follow up comment; we do not know which data set's (training or testing) performance results have been shown in the tables 3 to 7 and how and why the results in Table-4&6 differ than the ones shown in Table-9.

Reviewer #3: In this paper author proposed multi-class tasks for citation sentiments on imbalanced data.

The scheduler works in a hierarchical way as shown in Figure 1&2 and Algorithm 1. Jobs coming from multiple queues are split into tasks and sent to master schedulers.

While reasonably written, presentation has some problems as it shows multiple algorithms with no clear relation between them. For example, is algorithm 4 ever referred to and described in the text?

None of the ideas in this paper is new, e.g., as mentioned by author in references.

I believe that it is fair to say that authors built a variation of previously existing ideas and tried them in a fairly complex experimental setting. This is not to say that the implementation does not have anything new and different.

Graphics depicted are somewhat limited and straightforward, as only a handful of algorithms were used, and they are not necessarily aligned with the proposed scheduler in their goals.

Either Bracket ( or ) are missing/abused in many places, such as in equation (6), figure 2.

6. PLOS authors have the option to publish the peer review history of their article (what does this mean?). If published, this will include your full peer review and any attached files.

Reviewer #1: **Yes: **Mohammed Amin Almaiah

Reviewer #2: No

Reviewer #3: No

---

## [Author Response · Author response to Decision Letter 0]

18 Nov 2023

WE have added separate PDF file to address response to reviewers.

---

## [Decision Letter · Decision Letter 1]

27 Dec 2023

PONE-D-23-24549R1Scientific Text Citation Analysis Using CNN Features and Ensemble Learning ModelPLOS ONE

Dear Dr. Alnowaiser,

Thank you for submitting your manuscript to PLOS ONE. After careful consideration, we feel that it has merit but does not fully meet PLOS ONE’s publication criteria as it currently stands. Therefore, we invite you to submit a revised version of the manuscript that addresses the points raised during the review process.

We look forward to receiving your revised manuscript.

Kind regards,

Mohamed Hammad, Ph.D.

Academic Editor

PLOS ONE

Reviewers' comments:

Reviewer's Responses to Questions

**Comments to the Author**

1. If the authors have adequately addressed your comments raised in a previous round of review and you feel that this manuscript is now acceptable for publication, you may indicate that here to bypass the “Comments to the Author” section, enter your conflict of interest statement in the “Confidential to Editor” section, and submit your "Accept" recommendation.

Reviewer #2: All comments have been addressed

Reviewer #3: (No Response)

2. Is the manuscript technically sound, and do the data support the conclusions?

Reviewer #2: Partly

Reviewer #3: Yes

3. Has the statistical analysis been performed appropriately and rigorously? 

Reviewer #2: Yes

Reviewer #3: Yes

4. Have the authors made all data underlying the findings in their manuscript fully available?

Reviewer #2: Yes

Reviewer #3: Yes

5. Is the manuscript presented in an intelligible fashion and written in standard English?

Reviewer #2: Yes

Reviewer #3: Yes

6. Review Comments to the Author

Reviewer #2: It is a detailed comparative work yet I will raise some concerns as follows:

Novelty of this paper is not clear to me, anything has been implemented in the paper was already found on literature (eg a combination use of SMOTE with some ML algorithms or feature extraction with CNN despite in [44] a different set of embedding has been used, they still use the CNN concept for feature extraction..)

Logistic Regression and SVM are binary classifiers, methods section could have explained how they were covered to a multi class classifiers (eg one-vs-one, one-vs-all; which has been described only for SGC but not for LR or SVM)

No other implementation details provided on the materials (parameter or hyperparameter settings eg number of trees in RF, kernel type and values used in SVC etc)

On the line 298 (Page 8) it should be 'convolutional layer', not 'conventional'

I do not agree with the discussion/conclusion made regarding why the VC was performing poor in the case of using SMOTE (tables 4 and 6); 'if both classifiers have the same way of learning feature patters' as claimed by the author, then their common use would perform somewhere close to the individual use of these models (eg ~93% of accuracy) since they would generate similar probability scores for a given data point. Contrarily, I think these two models learn different (non-consensus) patterns for a given data point hence they agree lesser on the final probability value of VC..

Reviewer #3: Thanks for the opportunity to review the revised version of this paper. The authors have addressed most of the concerns, and I appreciate the authors’ efforts.

Comments:

Minor:

1. Figure 1 is not very clear in the printed version. The font size (e.g., the font size of some text in the figure) is a little small, which makes it a little difficult for the readers to clearly see the content in the figure. Considering that there is enough space, it would be better to enlarge the font size a little.

2. The presentation of the paper can be further improved. There are some typos in the paper. Here are some examples:

a. Page 6, line 235: “tree” could be changed to “Tree” for consistency.

b. Page 6, line 238: a space between “.” and “ETC” is missing.

c. Page 9, line 326: “figure 1” could be changed to “Figure 1”.

3. The format of the reference could be improved. For example, the format of references 2, 3, 8, 11 is a little different from that of other references. There are two punctuations “;” and “.” after the last word in the sentence.

4. This paper uses the SMOTE algorithm to handle class imbalance problem. Some other related works also use SMOTE algorithm to balance the dataset (e.g., the work focusing on extracting textual features of financial social media). Many feature selection methods are usually based on one chosen approach only without considering combining dissimilar feature selection approaches to enhance the performance. Sometimes hybrid feature selection can better enhance the performance. Probably the authors can consider it in the future work.

7. PLOS authors have the option to publish the peer review history of their article (what does this mean?). If published, this will include your full peer review and any attached files.

Reviewer #2: No

Reviewer #3: No

---

## [Author Response · Author response to Decision Letter 1]

20 Jan 2024

We have provided separate PDF file for response to reviewers.

---

## [Decision Letter · Decision Letter 2]

2 Apr 2024

Scientific Text Citation Analysis Using CNN Features and Ensemble Learning Model

PONE-D-23-24549R2

Dear Dr. Alnowaiser,

We’re pleased to inform you that your manuscript has been judged scientifically suitable for publication and will be formally accepted for publication once it meets all outstanding technical requirements.

Kind regards,

Mohamed Hammad, Ph.D.

Academic Editor

PLOS ONE

Additional Editor Comments (optional):

Reviewers' comments:

Reviewer's Responses to Questions

**Comments to the Author**

1. If the authors have adequately addressed your comments raised in a previous round of review and you feel that this manuscript is now acceptable for publication, you may indicate that here to bypass the “Comments to the Author” section, enter your conflict of interest statement in the “Confidential to Editor” section, and submit your "Accept" recommendation.

Reviewer #2: All comments have been addressed

2. Is the manuscript technically sound, and do the data support the conclusions?

Reviewer #2: Partly

3. Has the statistical analysis been performed appropriately and rigorously? 

Reviewer #2: N/A

4. Have the authors made all data underlying the findings in their manuscript fully available?

Reviewer #2: Yes

5. Is the manuscript presented in an intelligible fashion and written in standard English?

Reviewer #2: Yes

6. Review Comments to the Author

Reviewer #2: (No Response)

7. PLOS authors have the option to publish the peer review history of their article (what does this mean?). If published, this will include your full peer review and any attached files.

Reviewer #2: No

---

## [Editor Report · Acceptance letter]

14 May 2024

PONE-D-23-24549R2 

PLOS ONE

Dear Dr. Alnowaiser, 

I'm pleased to inform you that your manuscript has been deemed suitable for publication in PLOS ONE. Congratulations! Your manuscript is now being handed over to our production team.

Kind regards, 

on behalf of

Dr. Mohamed Hammad 

Academic Editor

PLOS ONE